# Tracking Missing Person in Large Crowd Gathering Using Intelligent Video Surveillance

**DOI:** 10.3390/s22145270

**Published:** 2022-07-14

**Authors:** Adnan Nadeem, Muhammad Ashraf, Nauman Qadeer, Kashif Rizwan, Amir Mehmood, Ali AlZahrani, Fazal Noor, Qammer H. Abbasi

**Affiliations:** 1Faculty of Computer and Information System, Islamic University of Madinah, Madinah 42351, Saudi Arabia; a.alzahrani@iu.edu.sa (A.A.); mfnoor@iu.edu.sa (F.N.); 2Department of Physics, Federal Urdu University of Arts, Science & Technology, Karachi 75300, Pakistan; m.ashraf@fuuast.edu.pk; 3Department of Computer Science, Federal Urdu University of Arts, Science & Technology, Islamabad 45570, Pakistan; nauman.qadeer@fuuast.edu.pk (N.Q.); kashifrizwan@fuuast.edu.pk (K.R.); 4Department of Computer Science and Information Technology, Sir Syed University of Engineering and Technology, Karachi 75300, Pakistan; amir.mehmood@zu.edu.pk; 5James Watt School of Engineering, University of Glasgow, Glasgow G12 8QQ, UK; qammer.abbasi@glasgow.ac.uk

**Keywords:** missing persons tracking, spatio-temporal features, intelligent video surveillance, large crowd gathering, faces detection, Viola Jones cascades fusion, facial recognition, unconstrained environment

## Abstract

Locating a missing child or elderly person in a large gathering through face recognition in videos is still challenging because of various dynamic factors. In this paper, we present an intelligent mechanism for tracking missing persons in an unconstrained large gathering scenario of Al-Nabawi Mosque, Madinah, KSA. The proposed mechanism in this paper is unique in two aspects. First, there are various proposals existing in the literature that deal with face detection and recognition in high-quality images of a large crowd but none of them tested tracking of a missing person in low resolution images of a large gathering scenario. Secondly, our proposed mechanism is unique in the sense that it employs four phases: (a) report missing person online through web and mobile app based on spatio-temporal features; (b) geo fence set estimation for reducing search space; (c) face detection using the fusion of Viola Jones cascades LBP, CART, and HAAR to optimize the results of the localization of face regions; and (d) face recognition to find a missing person based on the profile image of reported missing person. The overall results of our proposed intelligent tracking mechanism suggest good performance when tested on a challenging dataset of 2208 low resolution images of large crowd gathering.

## 1. Introduction

Tracking and locating a person automatically in an unconstrained large crowd gathering through face detection and recognition is still challenging. Face detection and recognition is challenging due to various dynamic factors such as low resolution, variable crowd distance from installed cameras, mobility of cameras and the crowd. In this paper, we propose an automatic tracking of the registered missing persons in a large Al-Nabawi mosque gathering scenario where millions of pilgrims gather to perform religious rituals in a covered area of approximately 73 acres.

The contribution in this paper is novel in two aspects. First, to the best of our knowledge, this is one of the few proposals for automatic tracking of the missing persons in a large gathering with low-resolution images. There are various proposals in the literature, which apply face recognition algorithms to large crowd images such as [1,2,3,4]; however, tracking a person in a large crowd with low-resolution images is rare. Several state-of-the-art deep learning algorithms are used for face recognition but with high-quality images such as [5]; however, our study of the related work suggested that they do not show good performance on low-resolution images of the unconstrained environment. To the best of our knowledge, dividing the surveillance premises into geofences and estimating set of geofences for probable presence of missing person is used for the first time in such works.Therefore, we believe our proposal in this paper contributes to the existing knowledge.

Second, our proposed mechanism is unique in terms of its methodology. In which we first efficiently reduce the search space to locate the missing person by applying the proposed geofence set estimation algorithm and then employ our face detector algorithm which fuses the three cascades of Viola Jones to optimize the number of localized face regions from each frame. Finally, the proposed mechanism applies the face recognition algorithm using the registered profile images of the missing persons to track them in the crowd.

We consider the unconstrained large gathering scenario of Al-Nabawi mosque, Madinah, Saudi Arabia. We first divided the total area of the mosque in 25 geofences and 20 cameras installed in this covered areas as our source of input data in the form of video sequences. Then, the system extracts frames of low quality images. We developed a mobile and web based system through which a head of the pilgrim’s family or the head of the pilgrims group can report his missing companion with time and location. Then, proposed Geo fence set estimation algorithm will result in suggested set of geofences where the missing person could be found. This will significantly reduce the search space and the system will start tracking from the videos of cameras installed in the suggested geo fences premises. Then, we apply our face detector algorithm which is a fusion of three Viola Jones cascades which produces high number of localized face regions with accuracy which is much higher than applying the viola Jones cascades individually. Then, face recognition algorithms is employed to find the missing person from the detected face regions based on the profile image of reported missing person. The prediction of proposed system in this paper has improved significantly from our previous work [6] where we just employed a single face detector algorithm. Secondly, it is an automated system as it first reduces the search space to increase the efficiency in terms of time and then employ our proposed face detector and recognition algorithm to track missing person in challenging unconstrained large gathering scenario with low resolution data.

The paper is structured as follows. Section 2 presents our brief related work of recent studies including gap analysis. Our proposed methodology is presented in Section 3 including algorithms and technical details. Then, we elaborate the implementation results of training and testing of our proposed methodology on the dataset in Section 4. Finally, the conclusion and future work is presented in Section 5.

## 2. Related Work

Facial recognition is critical for real-time applications of locating missing persons. Therefore, for our presented scenario it is the matter of an immense importance to identify and recognize human faces in a large crowd having unconstrained environment. Therefore, missing person identification could be attained to find vulnerable group of persons including elderly, children and people with disease (i.e., Dementia, Alzheimer, etc.). We now briefly present our review of recent literature related to the tracking of missing persons in the large crowd scenarios using face detection and recognition on video sequences.

According to Sanchez-Moreno, A.S. et al. [7], some deep neural networks techniques have recently been created to attain state-of-the-art performance on tracking of missing person through face detection and recognition problem. Their work is not for a densely populated environment. They employed the YOLO Face approach for face identification because of its high speed real time detector based on YOLO version 3. Secondly, for classification to recognize faces, a combination of FaceNet and a supervised learning technique, such as the support vector machine (SVM), is proposed. Their experiments are based on unconstrained datasets and show that YOLO-Face performs better when the face under analysis has partial occlusion and position fluctuations. Nonetheless, it can recognise little faces. The Honda/UCSD dataset is used to obtain an accuracy of more than 89.6 percent for face identification. Furthermore, the testing findings showed that the FaceNet+SVM model achieved an accuracy of 99.7 percent when utilizing the LFW dataset. FaceNet+KNN and FaceNet+RF score 99.5 percent and 85.1 percent, respectively, on the identical dataset, while FaceNet achieves 99.6 percent. Finally, when both the face detection and classification phases are active, the suggested system has a recognition accuracy of 99.1 percent and a run-time of 49 ms.

The early work published by Nadeem A. et al. [6] and Nadeem A. et al. [8] using an unique integration of face-recognition algorithms, which employs many recognition algorithms in parallel and combines their predictions using a soft voting mechanism, shown improved accuracy. Based on spatio-temporal constructs, this delivers a more sophisticated forecast. However, the technique was used on low-resolution cropped photos of recognized faces in order to discover missing people in the previously described difficult large crowd gathering. That was explored for scenarios involving enormous crowds at the Al-Nabawi mosque in Madinah. It is a highly unregulated environment with a data collection of low-resolution pictures collected from publicly recorded moving CCTV cameras. The proposed model first detects faces in real time from camera-captured photos, then applies face recognition algorithms with soft voting to get better prediction for identifying the missing persons. A tiny series of consecutive frames reveals the presence of a missing individual.

The method suggested by Li, W., and Siddique, A. A., [9] used the notion of face recognition by utilizing a pre-trained LBPH Face Recognizer to identify the individual in the acquired frame in conjunction with a drone mounted camera to capture the live video stream. An inbuilt Raspberry Pi module analyses the obtained video data, detecting the intended individual with an accuracy of 89.1%.

The authors Ullah, R. et al. [10] proposed a real-time framework for human face detection and recognition in CCTV images over a 40 K images with different environmental condition, background and occlusions. In addition, they performed a comparison analysis between different machine / deep learning algorithms such as decision trees, random forest, K-NN and CNN. The authors claimed that they have achieved 90% overall accuracy with minimum computing time using CNN.

As we noticed in Table 1, the authors in [7] applied state-of-art deep learning techniques for face detection and recognition using conversion of low resolution images to high-quality images, but the technique is not tested in low-resolution images from large gatherings. Moreover, literature in [1,2,3,4,5,11] shows work on recognizing people based on large crowd and low resolution image data, whereas the literature presented in [12] only depicts exploitation of large crowd data and in [13] research carried out only on low resolution data. However, emotional expression of human face have been found in [4] crowded environment showed happy faces are easily by identified. Finally, we found research in [14] that carried out identifying and tracking of pilgrims using CNN and Yolo v4 in unconstrained environment but used high resolution images data to identify smaller sized faces in the crowd.

By analyzing the state-of-the-art, we therefore state that no significant work found with human facial recognition and tracking of missing person based work on low resolution dataset in unconstrained and large crowded environments with above mentioned constraints. However, an ample amount of literature was found in the large crowd domain based on re-identification, tracking and crowd count, etc. Therefore, in this study, we presented our proposed mechanism considering the gap to find missing person by identification and recognition as well in a large crowded gathering of people with diverse age groups having fully unconstrained environment. In this regard, we used dataset built on the pre-processed frames extracted from publicly filmed CCTV videos in Al-Nabawi Mosque, Al Madinah, KSA.

## 3. Proposed Methodology

This research work is proposed for the automated tracking of reported missing person from live videos of unconstrained large gathering. The proposed mechanism is general but to prove the concept we consider large gathering scenarios of Al-Nabawi mosque (Madinah, KSA) where thousands of pilgrims daily visit. The probability of losing vulnerable companions such as a child or an older person in such large gatherings is high and their automated tracking, using intelligent video surveillance, is an extremely challenging task. The proposed work tries to mitigate this challenging task by dividing the experimented premises into geofences where each geofence is installed with particular cameras.

The proposed tracking mechanism is efficient as it reduces the search space by estimating the probabilistic region of a missing person through a novel geofence set estimation algorithm. This algorithm uses spatio-temporal information of a missing person reported by his/her group head through a mobile application. The query face image of a missing person is fetched from database and, afterwards, it is recognized in videos by cameras that are installed within output set of estimated geofences. This task is accomplished by applying face recognition algorithm on all detected faces which were detected in earlier stage by applying face detection algorithm on video frames. The main tasks of proposed methodology are depicted in Figure 1 and the details about all these tasks are given in following subsections.

### 3.1. Reporting Missing Person

It is a general practice that pilgrims in Al-Nabawi mosque move in the form of groups. Sometimes each family has a separate group headed by a family member, as shown in Figure 2a, and sometimes it consists of people from different families and headed by a group leader as shown in Figure 2b.

The missing person reporting is conducted by group leader through a mobile application. The reporting includes selecting the missing person from the list of his/her group members. This information is accompanied by the approximate missing time and location (in terms of geofence). This information is passed on to geofence set estimation algorithm given in next section.

### 3.2. Geofence Set Estimation

The perimeter of Al-Nabawi mosque (including the courtyard) is calculated as 2.166 km. We further divided this premises into a 5 × 5 matrix of square sized geofences. The partitions are shown in Figure 3a and the exact dimensions are shown in Figure 3b.

The whole premise is covered by 20 surveillance cameras that are installed inside the boundary of Al-Nabawi mosque. Each surveillance camera covers a particular set of geofences. The missing person reporting includes spatio-temporal features of missing event. It includes geo-location of missing person that is approximated by mobile based location of group head and it also includes the estimated time laps (in minutes) since the person is missed. Therefore, based upon this information, a set of geofences is obtained by applying geofence set estimation algorithm. This algorithm defines several crowd levels based upon the automated counting score of people. Then, based upon that crowd level score, the maximum possible distance, covered by missing person, is calculated around all four possible directions and finally a set of geofences is calculated where that person can be found. The algorithm’s output reduces the search space and hence the missing person is tracked only in videos of those cameras that are installed within the output set of geofences. The geofence set estimation is given in Algorithm 1.
**Algorithm 1** Geo-Fence Set Estimation1:*Begin***input:*****t*** (estimated time laps, in minutes, since person missed), ***l*** (mobile based location of group head)**output:*****G*** (set of geo-fences / Range for probability of presence of the missing person)2:Derive geo-fence $G_{ij}$ of group head (i.e., reporting person) based upon his reporting mobile’s location *l*3:Calculate crowd level (i.e., $CL_{ij}$) in all geo-fences $G_{ij}\left(i,j=1\ldots 5\right)$ based upon total sum $\left(S_{ij}\right)$ of automated counting score of persons in placed n camera images in that geo-fence premises. categorized crowd score levels as per following rule:$\;\;\;\;\;\;\;\;\;\;CL_{ij}=Round\left(S_{ij}/60\right)$4:$x=(t\times CL_{ij})/n$5:A={},B={},C={},D={}6:**if**((i+x)>5)**then**7:   A=((i+x−5)×110) meters outside mosque premises8:**end if**// (i.e., calculating vertically down from current geo-fence)9:**if** 
((i−x)<1)
 **then**10:   B=(|i−x|×110) meters outside mosque premises11:**end if**// (i.e., calculating vertically up from current geo-fence)12:**if**((j+x)>5)**then**13:   A=((j+x−5)×110) meters outside mosque premises14:**end if**// (i.e., calculating horizontally right from current geo-fence)15:**if** 
((j−x)<1)
 **then**16:   B=(|j−x|×110) meters outside mosque premises17:**end if**// (i.e., calculating horizontally left from current geo-fence)18:**$G=G_{ij}⋃\sum_{a=i}^{i+x}\sum_{b=j+1}^{j+x}G_{ab}⋃\sum_{a=i}^{i+x}\sum_{b=j-x}^{j-1}G_{ab}⋃\sum_{a=i}^{i-1}\sum_{b=j+1}^{j+x}G_{ab}⋃\sum_{a=i}^{i-1}\sum_{b=j-x}^{j-1}G_{ab}$**19:**if** 
(A={}) and (B={}) and (C={}) and (D={})
 **then**20:   exit21:**else**22:   i.$\;\;\;G=G⋂\{\forall G_{ab}|\left(1\leq a\;\leq 5\right),\left(1\leq b\leq 5\right)\}$   ii.   G=G⋃A⋃B⋃C⋃D
23:**end if**24:*end*

In Algorithm 1 $CL_{ij}$ is monotonically increasing function and directly proportional to $S_{ij}$ value. This rule is based upon geo-fence length and width (which is nearly equal to 110 m as geo-fence is nearly square shaped) and 0.5 m/s is the observed average walking speed of person when the premises is nearly vacant (which is estimated as under 60–70 persons) and then walking speed reduced as geo-fence premises gets populated. As per the observation, the average speed of walking person reduced to nearly half as it gets double populated (i.e., 90–120 persons in a geo-fence) then further reduced to one-third when it gets populated nearly 150–200 persons and so on.

### 3.3. Faces Detection in Video Frames

The faces are detected at frame level, whereas the video streams of only those cameras are examined which are installed within the geofences mentioned in output set of estimation algorithm described earlier. A tracking workflow that examines the video streams is presented in Algorithm 2, which is the improved version of our previously proposed tracking workflow in [6]. It samples every 10th frame and detects the face regions on that frame. There exists several face detectors, but no one is capable of detecting all the faces in given image correctly. Therefore, a sampled frame is simultaneously fed to three established face detectors called the Cascaded CART, the Cascaded Haar and the Cascaded LBP face detector, then output from these detectors is merged to improve the face detection process. A new face fusion technique is proposed in Algorithm 3, which not only controls the merger of detected overlapping faces, but also maintain the bounding box for updated face region. The fusion strategy increases the face count at frames level, which may also increase a person face count in temporal domain. Therefore, it will enhance the missing person tracking by reducing the negative errors.
**Algorithm 2** Tracking workflow1:// create face detector objects2:$FaceDetect^{cart}\leftarrow vision.CascadeObjectDetector\left(\mathrm{Frontal}\;\mathrm{Face}\;\mathrm{CART}\right)$3:$FaceDetect^{haar}\leftarrow vision.CascadeObjectDetector\left(\mathrm{Frontal}\;\mathrm{Face}\;\mathrm{Haar}\right)$4:$FaceDetect^{lbp}\leftarrow vision.CascadeObjectDetector\left(\mathrm{Frontal}\;\mathrm{Face}\;\mathrm{LBP}\right)$5:**while** 
on
 **do**6:   $fr^{t}\leftarrow camera$// get video frame *t*…7:   $BB^{cart}\leftarrow step\left(FaceDetect^{cart},fr^{t}\right);$ // face detection by CART…Where BB defines set of bounding boxes8:   $BB^{haar}\leftarrow step\left(FaceDetect^{haar},fr^{t}\right);$ // face detection by Haar…9:   $BB^{lbp}\leftarrow step\left(FaceDetect^{lbp},fr^{t}\right);$ // face detection by LBP…10:   $BB\leftarrow \mathrm{Face}\;\mathrm{Fusion}\left(BB^{cart},BB^{haar},BB^{lbp}\right)$11:   **for** ∀b∈BB **do**12:     $f\leftarrow fr^{t}\left(b\right)$ // crop the face region13:     $f_{er}\leftarrow imresize\left(f,interpolation,\left[50,50\right]\right);$14:     **for** ∀j∈Alogs **do**15:        ${\left(ID,Score\right)}_{j}\leftarrow algo_{j}\left(f_{er}\right);$// for *j*th algorithm16:     **end for**17:     $\left(ID,Score\right)\leftarrow Voting(\left[{\left(ID,Score\right)}_{1},\ldots ,{\left(ID,Score\right)}_{5}\right])$18:   **end for**19:   $Tracks\leftarrow Tracking(IDs^{t},IDs^{t-1},IDs^{t-2},IDs^{t-3})$20:   // go for next frame21:**end while**

**Algorithm 3** Proposed Face fusion
1:**for** 
$\forall c\in BB^{cart}$
**do**2:   **for** $\forall h\in BB^{haar}$ **do**3:     $a_{iou}\leftarrow \left[\left(c\cap h\right)/\left(c\cup h\right)\right]$4:     **if** $a_{iou}>0.50$ **then**5:        $BB^{cart}\leftarrow \left(c\cap h\right)$ // over write the box *c* in $BB^{cart}$6:        $BB^{haar}\leftarrow BB^{haar}-h$ // cut the box *h* from $BB^{haar}$7:     **end if**8:   **end for**9:
**end for**
10:

$BB^{fusion}\leftarrow BB^{cart}\cup BB^{haar}$

11:**for** 
$\forall f\in BB^{fusion}$
**do**12:   **for** $\forall l\in BB^{lbp}$ **do**13:     $a_{iou}\leftarrow \left[\left(f\cap l\right)/\left(f\cup l\right)\right]$14:     **if** $a_{iou}>0.50$ **then**15:        $BB^{fusion}\leftarrow \left(f\cap l\right)$ // over write the box *f* in $BB^{fusion}$16:        $BB^{lbp}\leftarrow BB^{lbp}-l$ // cut the box *l* from $BB^{lbp}$17:     **end if**18:   **end for**19:
**end for**
20:

$BB^{fusion}\leftarrow BB^{fusion}\cup BB^{lbp}$




The face regions detected on a sampled frame are shown in Figure 4, where face regions detected by individual face detectors can be observed clearly. Figure 5 shows the comparative analysis of detected faces, where overlapping and non-overlapping face regions can be observed easily. The great extent of overlap recommend fusing these regions to a single face region, where additional efforts may be required to adjust the bounding box over updated face region. The process not only adjust the bounding boxes but also increases the face count on sampled frame.

### 3.4. Face Recognition

All the face regions detected on a sampled frame are cropped, enhanced and resized to a size of 50 × 50. Then, every face image is simultaneously fed to five recognition algorithms, where each algorithm provides an (ID, Score) pair for that face image. All the five algorithms may or may not predict the same identification result for input face, therefore obtained (ID, Score) pairs are fed to a soft-voting algorithm that produces a mature identification result for input face. The details about the voting scheme can be seen in our previously published paper [6]. An example of recognizing the detected face regions is presented in Figure 6a, where predicted identity of every face region is labeled on box and the associated score is illustrated by bounding box color. The score-color scheme is completely in accordance with [6], where NM-2 stands for no match recommended and NM-1 indicates no match suggested due to the confusion. Face regions with white boxes show no identity, and a tag of NM-1 or NM-2 is mentioned on every white box, which means this face region does not match to any personnel stored in database. The actual identification of faces on sampled frame is presented in Figure 6b. Only 15 faces were detected on sampled frame, where 9 faces find a proper match in database, while 6 faces did not find any valid match. The predicted identity of 9 faces on sampled frame can be confirmed from Figure 6b.

### 3.5. Missing Person Tracking

The missing persons are tracked in all cameras installed in recommended geofences. For example the tracking of subjects ID-47, 51 and 53 in three different camera views are shown in Figure 7a–c given below.

The presence of every identified personnel in video sequence is recorded temporally. The tracking of above 12 personnel is presented in Figure 8, where actual and predicted presence of 12 personnel is presented with different color. As the subject in video sequence is free to move his face leftward, rightward or downward, the presence may not be recorded at every frame correctly, and the ID track looks rough in temporal domain. To resolve this issue the IDs tracks are smoothed along time domain, which improves the tracking of missing personnel significantly. The roughness of an ID track is minimized by holding the presence record of that ID over multiple consecutive past frames.

The proposed technique of smoothing the presence track over temporal domain is presented in Figure 9.

The smoothed presence tracks of 12 personnel are presented in Figure 10, where false positive and the false negative presence of some of the 12 personnel can be seen easily.

## 4. Results

The experiments were conducted on a large gathering images dataset. These images were obtained through short videos captured by 20 installed surveillance cameras inside Al-Nabawi mosque as shown in Figure 11.

It consists of 2208 sampled raw video frames, processed face images and the presence tracks of 188 subjects. In the following subsections, the experimentation results for training and testing on our dataset are presented separately for face detection, face recognition and tracking.

### 4.1. Face Detection

Detecting faces in sampled frames is the first and important step that significantly affects the subsequent processes. There exist several face detectors but Viola Jones is the most frequently used face detector as it can quickly and accurately detect faces in the image. Although it shows good performance, but still some of the face regions are missed by the algorithm. Therefore, first using the cascaded face detectors of CART, HAAR and LBP in parallel and then fusing their output was proposed, which results in more detected faces than individual detection algorithms. The video sequence of 2208 sampled frames was fed to the system and the total number of faces detected over this sequence is presented in Figure 12, where Cascade HAAR detects a total of 7316 faces, Cascade CART a total of 6131 faces, Cascade LBP a total of 3317 faces and their fusion detects a total of 10235 faces on entire video sequence. The frame level detection counts are presented in Figure 13, where faces counts of three cascades fusion are better than individual cascaded algorithms. The qualitative appearance of detected faces is presented in methodology section, where face boxes representing the entire face region looks good. Since every detected face covers the entire face region, and the total number of faces counts increases, it will definitely support and enhance the subsequent recognition and tracking processes.

### 4.2. Face Recognition

Face recognition at every sampled frame plays a significant role in tracking the personnel in entire video sequence. The individual recognition algorithm may perform poorly, and results an incorrect identification, therefore input faces were fed to five recognition algorithms simultaneously and then resultant identifications were fed to a soft-voting algorithm to mature the input face identification. The recognition process is completely in accordance with [6] algorithm. Since face detection was executed by three face detectors, the matured recognition of detected faces for every face detector is presented in Figure 14, where recognition results over cascade CART and cascade fusion are better than other two detectors. The faces detected on sampled frame were matched to the stored faces in database and identification for every detected face was determined. Few of the faces did not find any match and were tagged “(NM-1 or NM-2)”, while remaining faces found a correct match in database. The tag of NM-1 stands for “No match suggested due to confusion,” and every algorithm assigns this tag to a face if it finds a little match for that face region, on the other hand the tag of NM-2 stands for “No match recommended,” and every algorithm assigns this tag to a face only if it finds a very little match for that face region. All the faces detected by CART on sampled frame were identified correctly. The stored reference faces, which can be found on the current sample frame are presented in Figure 6. The predicted identification identifications for most faces in sampled frame are exactly the same as mentioned in database.

### 4.3. Missing Personnel Tracking

The main objective of proposed work is to find an efficient tracking methodology for missing personnel, which definitely depend on face detection and recognition results. All the recognition algorithms were fine tuned to perform their optimal, the tracking results of each recognition algorithm against every detector is presented in following figures. The tracking results for PCA are presented in Figure 15, where precision and recall curves present the tracking analysis, the fine-tuned point is highlighted by plotting a circle on drawn curves. To further elaborate the fine tuning process, f1-score and accuracies are plotted against the tuning parameter, and fine-tuned points are highlighted by plotting a circle on f1-score curves. F1-score is an evaluation measure that finds a balance point over precision recall curves for optimal performance, it does not consider the faces which finds no match in recognition process and plays an important role where false positive and false negative errors have different impact. Accuracy curves are also plotted to evaluate the tracking performance. It considers true positive, false positive, false negative and true negative scores, and here considers the faces which find no match in stored database. Since we maintain the presence records of personnel stored in missing personnel database, and mainly focus ourselves over their tracking, the f1-score is given more importance than accuracy measure. Therefore we found the optimal point over f1-score curves and plotted the performance evaluation for those points. According to the evaluation measures presented in Figure 16, Figure 17, Figure 18, Figure 19 and Figure 20, tracking performance for cascades fusion is better than individual face detecting algorithms, which support our claim of using more than one detectors and then fusing their outputs for improving the face detection rate.

The overall f1-score and accuracy rate of cascade fusion is 67.1083% and 72.4924% (before smoothing) and 71.644% and 75.9249% (after smoothing) respectively.

## 5. Conclusions and Future Work

Large crowd management poses various challenges including tracking or locating a missing person and connecting him/her with his/her head of the family/group. In this paper, we present our work from a funded research project related to the automatic tracking of a missing person in an unconstrained large gathering scenario. We proposed a geofence set estimation to reduce the search space for finding registered missing persons. We first tested three Viola Jones cascades—CART, HAAR and LBP—individually on our unconstrained large gathering dataset for localization of face images. Then, to optimize the results of face detection, we proposed the fusion of these cascades which results in improving both the number of detected faces and their accuracy. This has subsequently helped in better face recognition and identification of the missing person. This work is limited to face recognition for tracking of missing person in videos of large crowd gathering scenarios. In order to cover other dimensions such as detecting missing person when his/her face is hidden, more research is required in other research fields such as “gait recognition”, “person re-identification” and “tracking using wearable devices” that are the part of our planned future work.

## Figures and Tables

**Figure 1 sensors-22-05270-f001:**
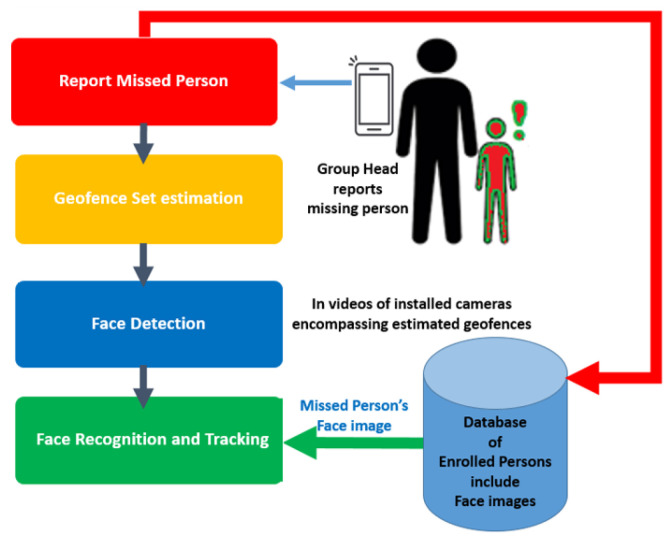
Main phases in the proposed methodology.

**Figure 2 sensors-22-05270-f002:**
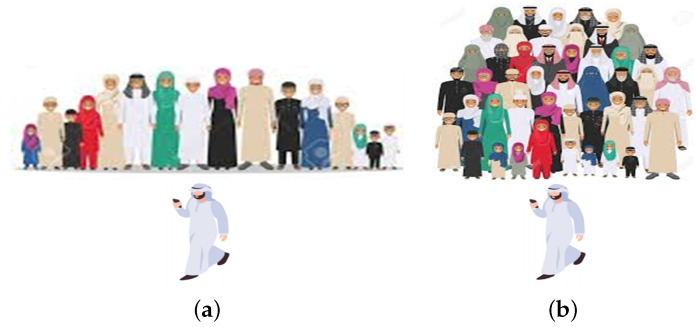
Groups with head/leader. (**a**) Family with head; (**b**) Group with leader.

**Figure 3 sensors-22-05270-f003:**
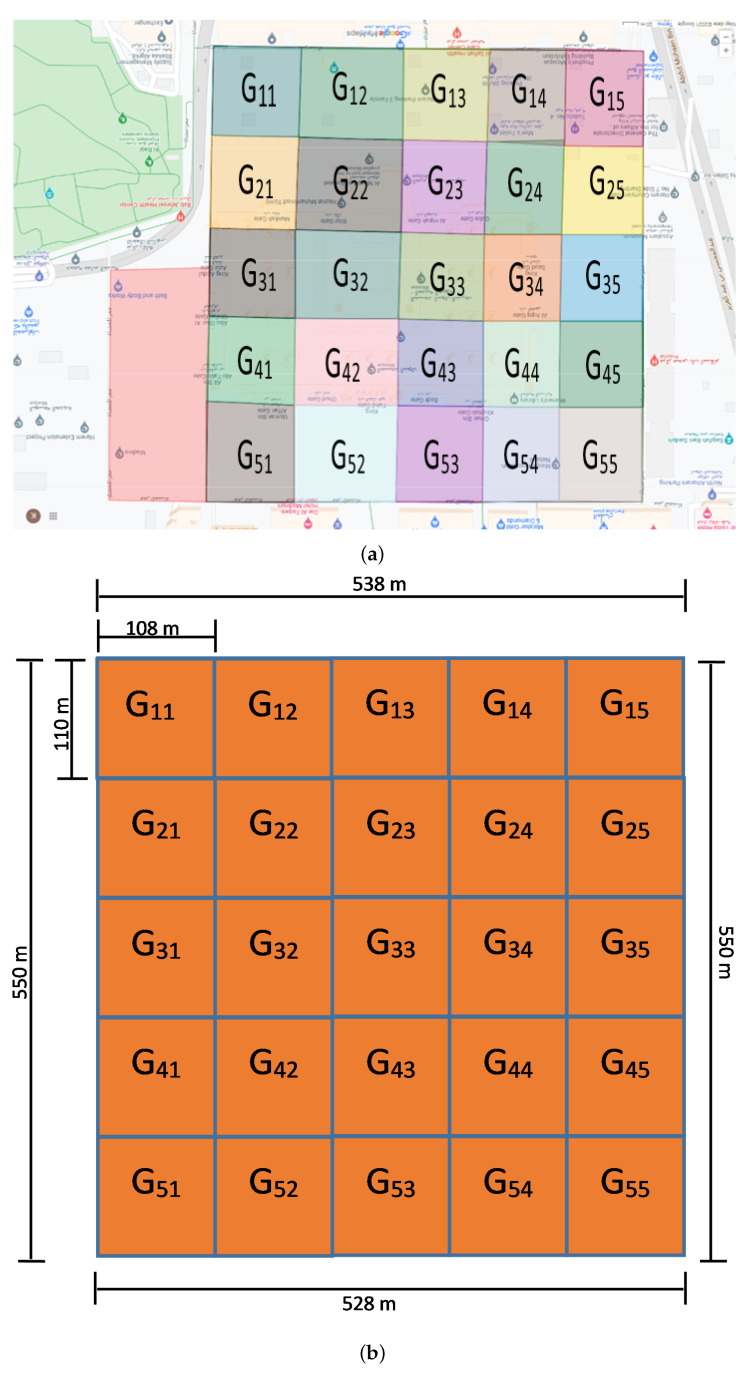
Geofences in Al-Nabawi mosque. (**a**) Geofences for Haram An-Nabavi; (**b**) Dimensions of geofences for Haram An-Nabavi.

**Figure 4 sensors-22-05270-f004:**
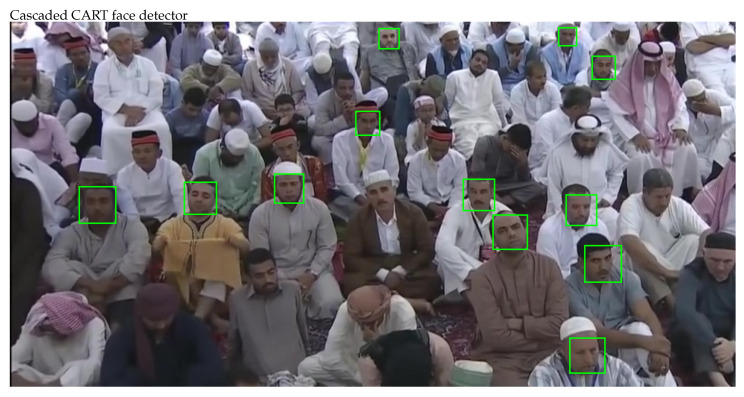
Face detection by three cascaded face detectors.

**Figure 5 sensors-22-05270-f005:**
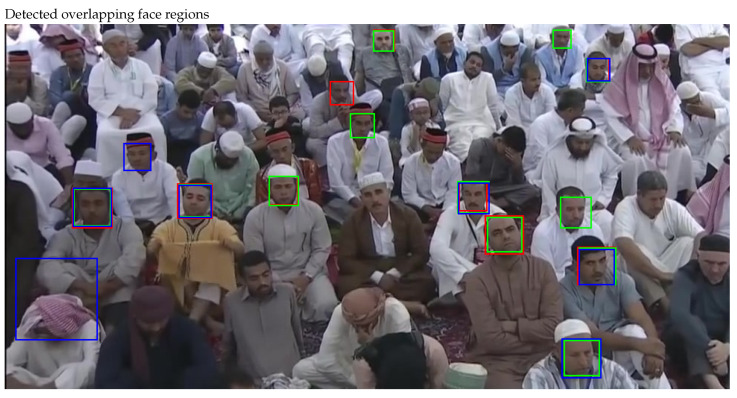
Cascades fusion by overlapping face regions (detected by three cascades).

**Figure 6 sensors-22-05270-f006:**
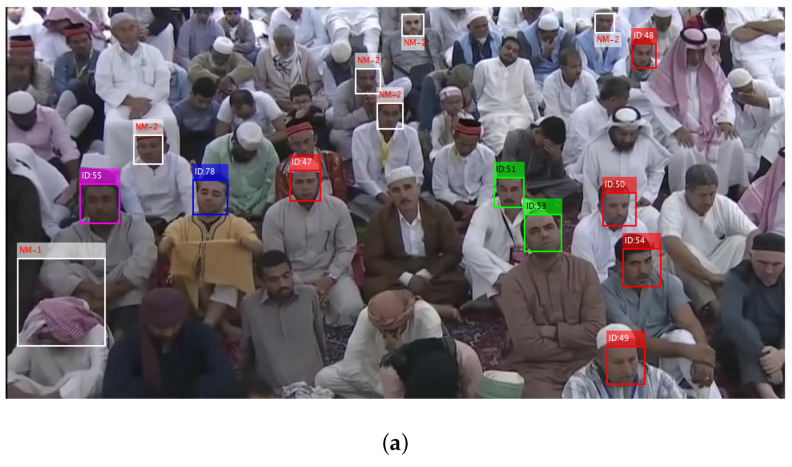
Recognized faces on sampled frame. (**a**) Predicted identification; (**b**) Actual identification.

**Figure 7 sensors-22-05270-f007:**
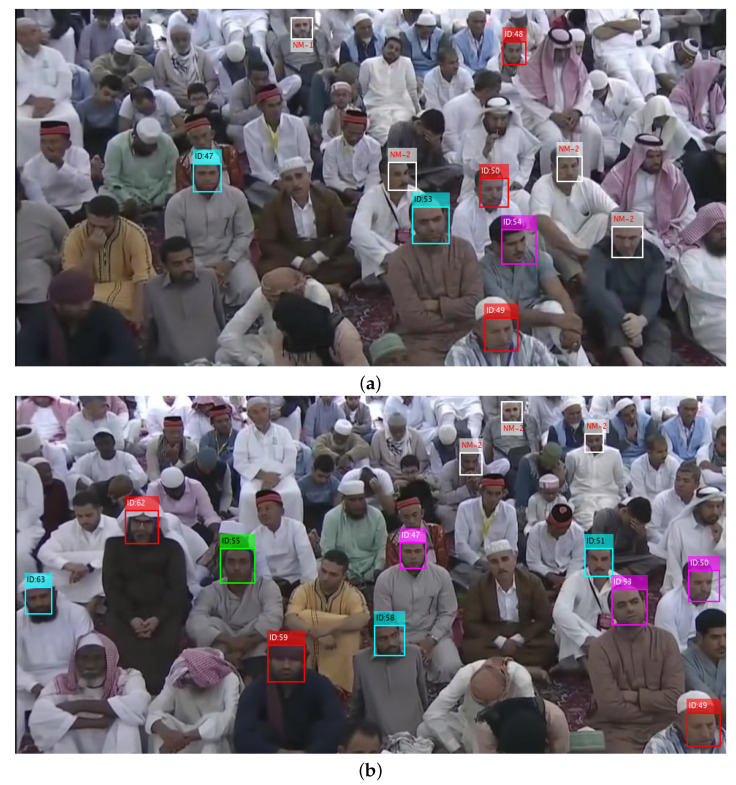
Tracking of subjects ID-47, 51 and 53 in three different camera views. (**a**) Camera view 1; (**b**) Camera view 2; (**c**) Camera view 3.

**Figure 8 sensors-22-05270-f008:**
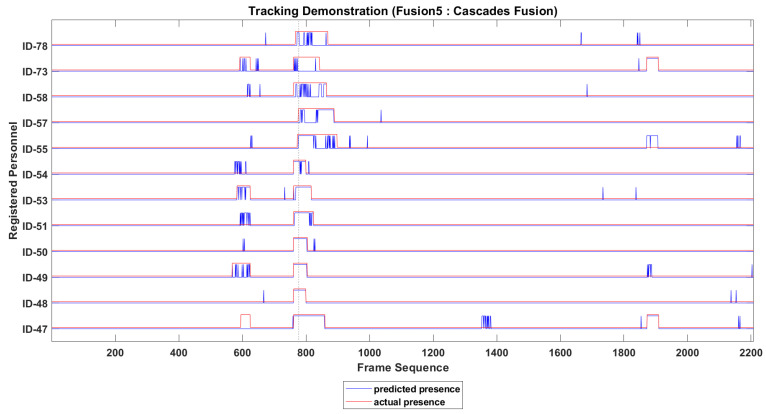
IDs tracking before smoothing over time domain.

**Figure 9 sensors-22-05270-f009:**
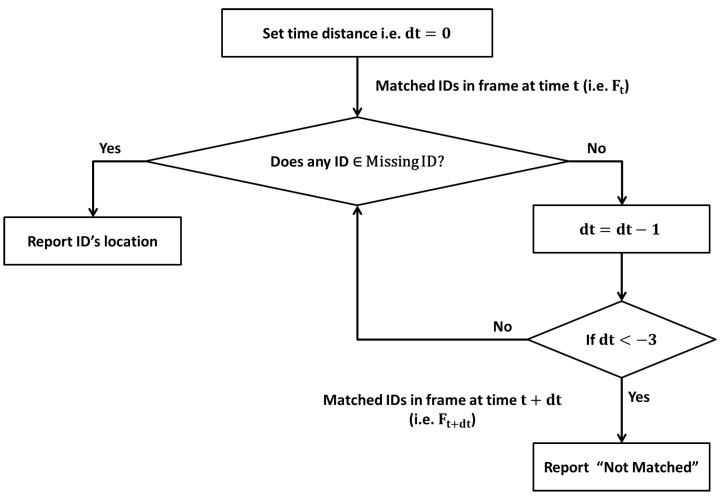
Smoothing operation to minimize irregularity of matched person presenc.

**Figure 10 sensors-22-05270-f010:**
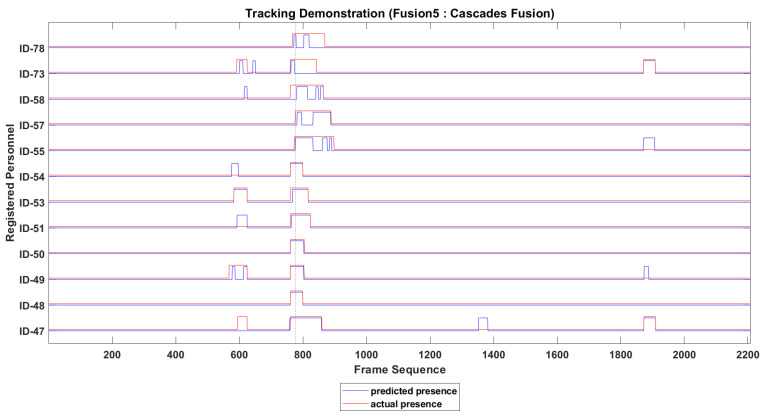
IDs tracking after smoothing over time domain.

**Figure 11 sensors-22-05270-f011:**
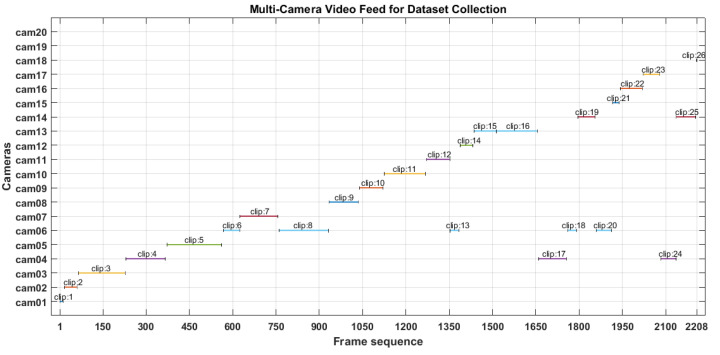
Multi-Camera video feed.

**Figure 12 sensors-22-05270-f012:**
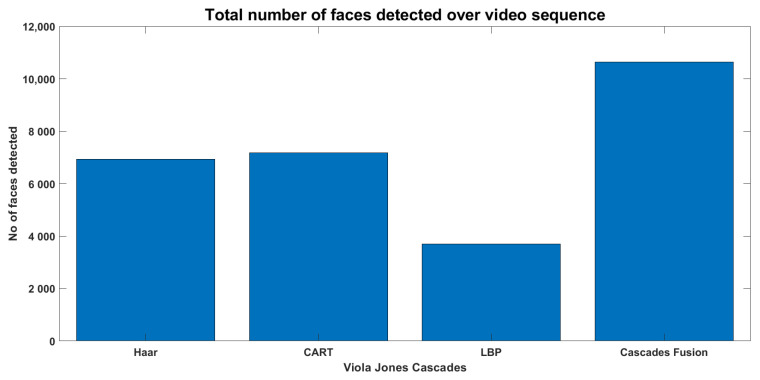
Detected faces counts on entire video sequence.

**Figure 13 sensors-22-05270-f013:**
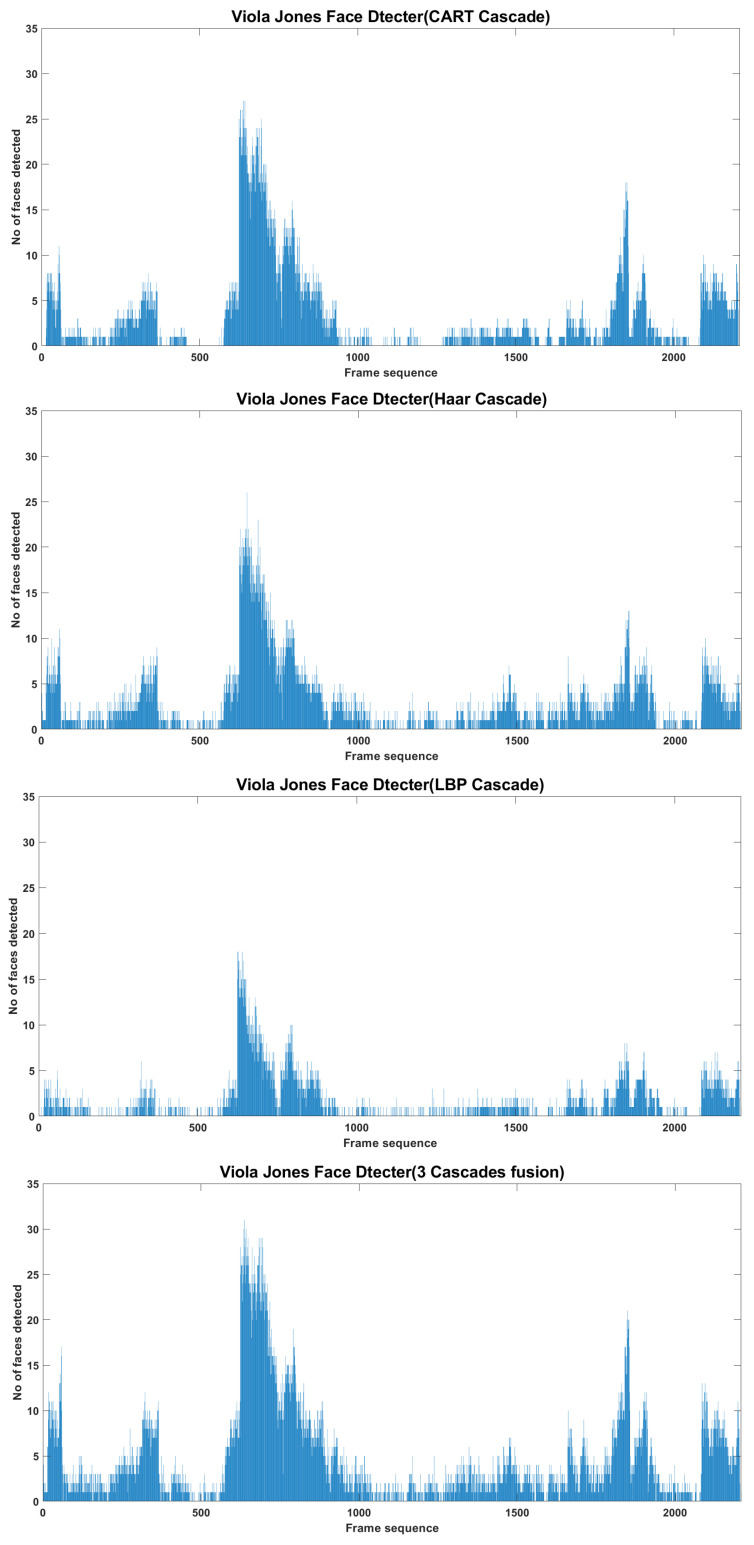
Detected faces counts on individual frames.

**Figure 14 sensors-22-05270-f014:**
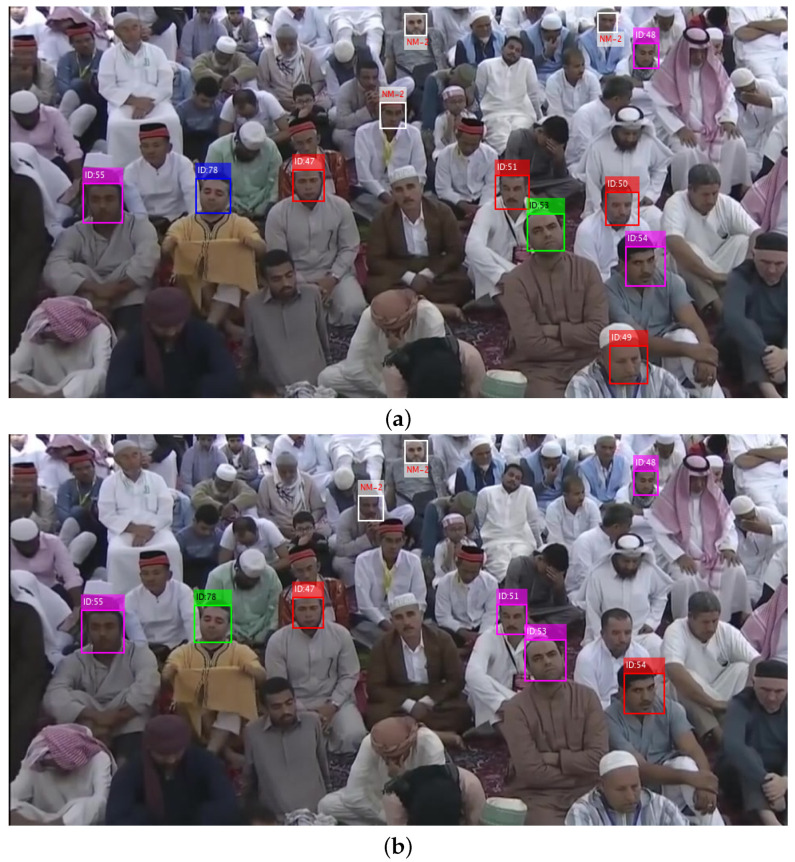
Facial Recognition using CART, HAAR, LBP and Cascade fusion. (**a**) Predicted identification for CART faces; (**b**) Predicted identification for Haar faces; (**c**) Predicted identification for LBP faces; (**d**) Predicted identification for Cascades fusion.

**Figure 15 sensors-22-05270-f015:**
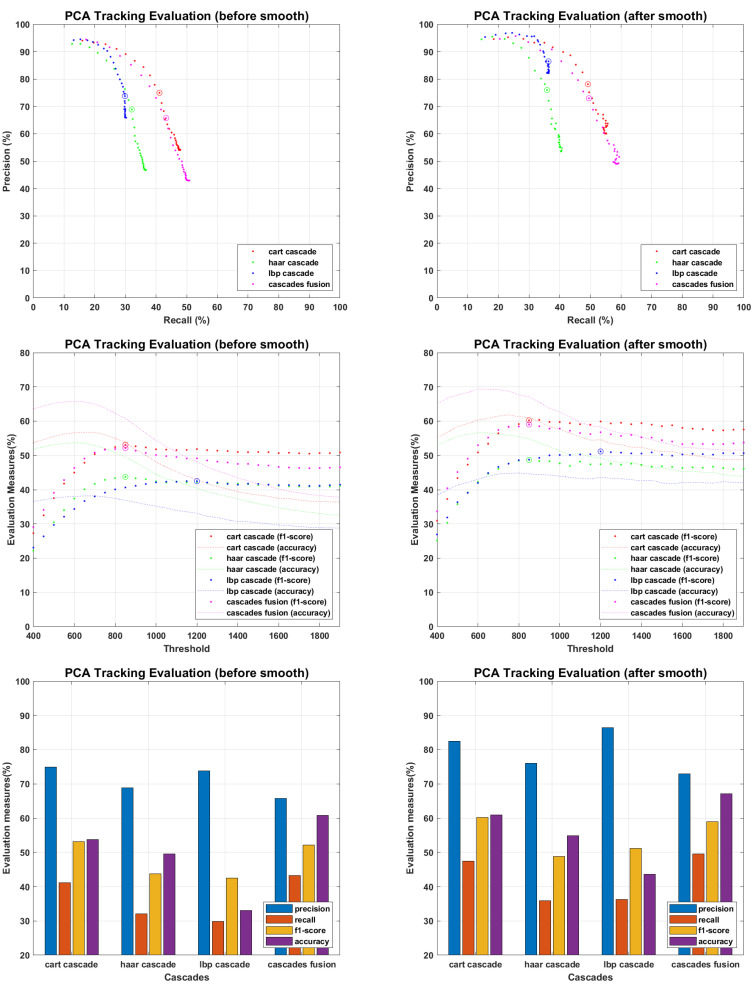
Tracking Evaluation for PCA.

**Figure 16 sensors-22-05270-f016:**
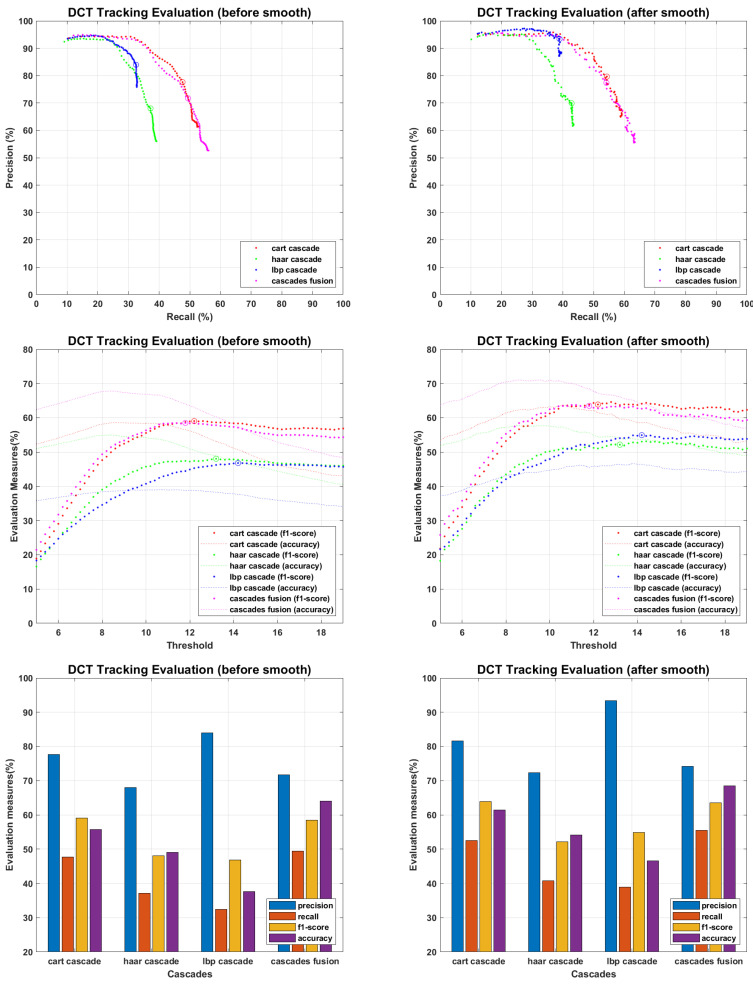
Tracking Evaluation for DCT.

**Figure 17 sensors-22-05270-f017:**
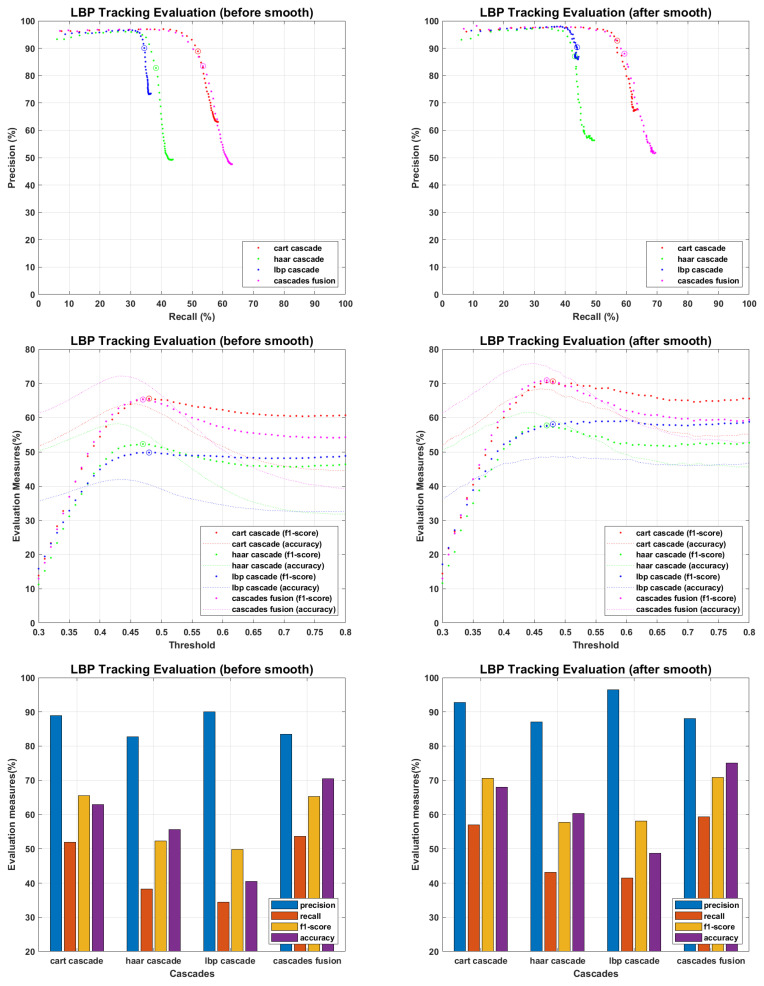
Tracking Evaluation for LBP.

**Figure 18 sensors-22-05270-f018:**
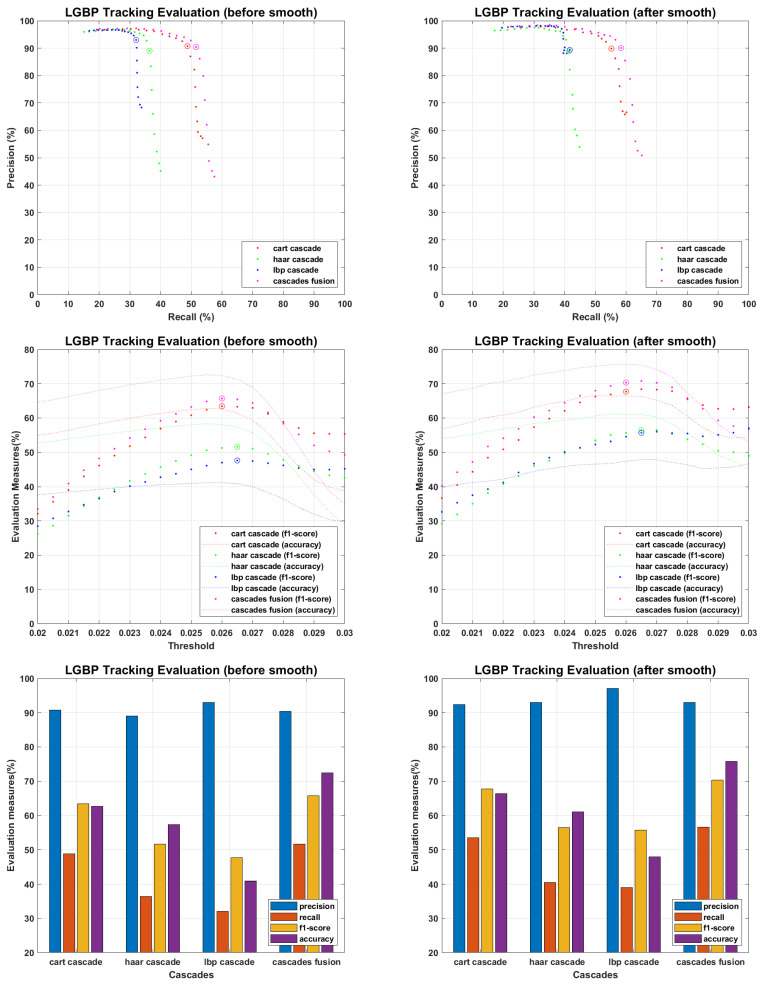
Tracking Evaluation for LGBP.

**Figure 19 sensors-22-05270-f019:**
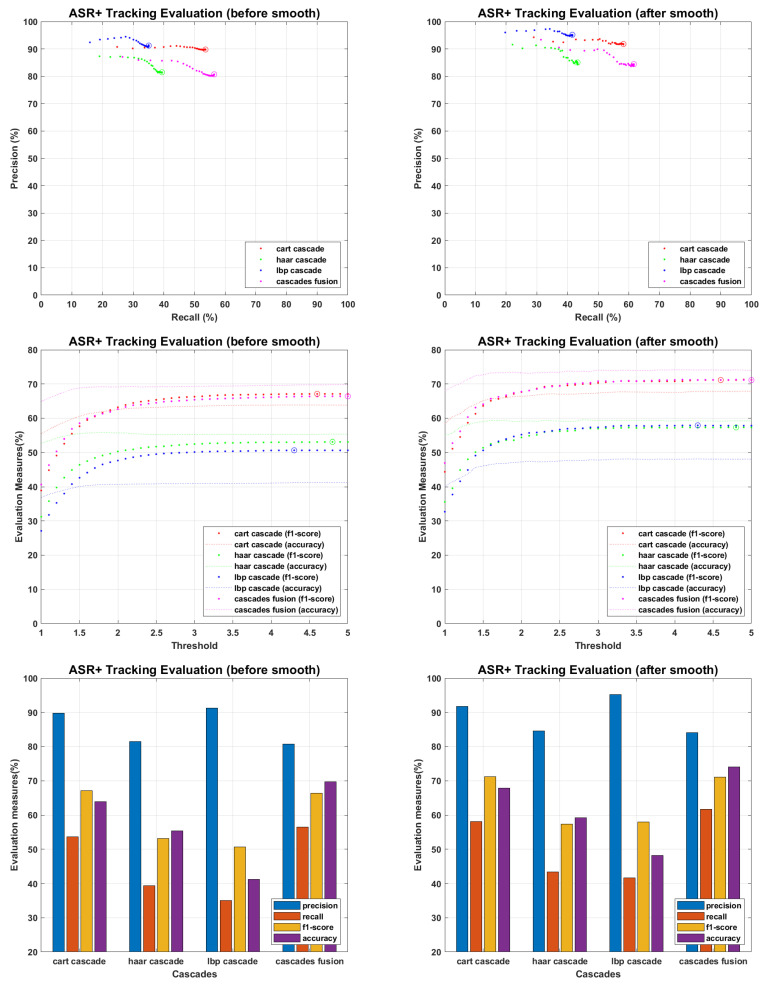
Tracking Evaluation for ASR+.

**Figure 20 sensors-22-05270-f020:**
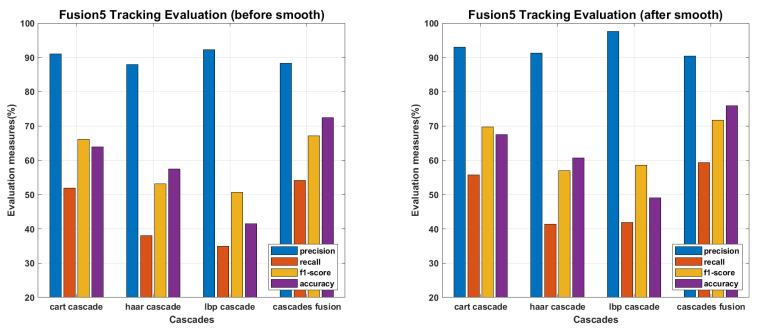
Tracking Evaluation for fusion of all recognition cascades through soft-voting scheme.

**Table 1 sensors-22-05270-t001:** Comparison of various parameters used in the problem domain.

Reference	Low Resolution Image Data	Huge Crowd Environment	Unconstrained Environment	Identifying and Recognizing for Tracking Persons	Used Machine Learning Algorithms for Detection (D)/ Recognition (R)	Fusion of Face Detection (D) and Recognition (R) Algorithms
[7]	Yes	Yes	Yes	No	FaceNet+SVM (D)/ YOLO v3 (R)	No
[6,8]	Yes	Yes	Yes	No	Viola Jones(D)/ PCA, DCT, LGBP, LBP, ASR+ (R)	Yes (R)
[9]	Yes	Yes	Yes	No	LBP(RH )	No
[1,2,3,4,5]	Yes	Yes	Yes	No	Well known algorithms for (D) & (R)	No
[14]	No	Yes	Yes	Yes	CNN (D) / YOLO v4 (R)	No
[10]	No	Yes	Yes	Yes	PCA, CNN (D) / DT, RF, KNN, CNN (R)	No
Proposed	Yes	Yes	Yes	Yes	LBP, CAR, HAAR (D) / PCA, DCT, LGBP, LBP, ASR+ (R)	Yes (D)&( R)

## Data Availability

The dataset will be available for further research in future.

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
