# Peer review of "Tracking Missing Person in Large Crowd Gathering Using Intelligent Video Surveillance"

_sensors, 2022, doi:10.3390/s22145270_

Round 1
Reviewer 1 Report
Locating a missing child or elderly person in a large gathering through face recognition in videos is still a challenging job. This study provides a promising method for tracking missing persons in an unconstrained large gathering scenario of Al Nabvi Mosque, Madinah, KSA. The work is interesting and readable. However, there is need to address following revisions/concerns before final publication.
1. How Geofence set estimation is estimated. Why this is required in missing person identification?
2. One major concern about feasibility of face detections with covered faces especially with women/girls. Is it possible? Can you work with IRIS in this scenario.
3. Why algorithm for geo-fence estimation is presented as figure in Fig.3. Write accordingly.
4. The authors should include and discuss other related deep learning architectures utilized in other application for a broader vision. Besides, authors should cite more related work, e.g.,
[1] Chattopadhyay, A., & Maitra, M. (2022). MRI-based Brain Tumor Image Detection Using CNN based Deep Learning Method. Neuroscience Informatics, 100060.
[2] Tiwari, S., & Jain, A. (2022). A lightweight capsule network architecture for detection of COVID‐19 from lung CT scans. International Journal of Imaging Systems and Technology, 32(2), 419-434.
5. Clearly mention the research contributions in introduction section.
6. Why face fusion is required as proposed in algorithm 2.
7. Improve the quality of Fig. 7-10.
7. Compare the proposed work with other state-of-the-art works in the same field.
Author Response
Reviewers Comments, Authors’ Reponses and Actions
Reviewer # 1
Reviewer#1, Comment # 1: Locating a missing child or elderly person in a large gathering through face recognition in videos is still a challenging job. This study provides a promising method for tracking missing persons in an unconstrained large gathering scenario of Al Nabvi Mosque, Madinah, KSA. The work is interesting and readable. However, there is need to address following revisions/concerns before final publication. Those suggestions are following:
Author response: We are thankful to the reviewer for acknowledging the potential our presented work.
.
_______________________________________________________________________________________
Reviewer#1, Comment # 2:
How Geofence set estimation is estimated. Why this is required in missing person identification?
Author response: This work is applied for automated video surveillance of large crowd gathering premises where many cameras are installed and each camera is covering a particular part of the premises. We have already explained in the manuscript that surveillance premises is equally partitioned into square sized geofences (kindly see Figure-2 and its explanation mentioned in manuscript). Furthermore, our proposed mechanism is specially designed for large crowd gathering scenarios where people move in the form of groups and hence absence of any group member is immediately reported by group head (though mobile application) thus specifying the estimated time laps (in minutes) and approximate missed location of missing person. So based upon this information, a set of geofences is obtained by applying Algorithm-1 (given at page 7 of manuscript). This algorithm defines several crowd levels based upon the automated counting score of people. Then based upon that crowd level score, the maximum possible distance, covered by missing person, is calculated around all four possible directions and finally a set of geofences is calculated where that person can be found. So in the way the search space is reduced and missing person is traced only in videos of cameras that are installed within those geofences.
Author action: As per response, we have added further clarification related to geofence estimation in section 3.2 (first paragraph on page 6 and 7).
_______________________________________________________________________________________
Reviewer#1, Comment # 3: One major concern about feasibility of face detections with covered faces especially with women/girls. Is it possible? Can you work with IRIS in this scenario.
Author response: This work is solely focus on utilizing the existing infrastructure of installed cameras in Al Nabawi Mosque, therefore we employed face detection and recognition for tracking. However, we have noted the suggestions of the reviewer and will IRIS based detection in future.
Reviewer#1, Comment # 4: Why algorithm for geo-fence estimation is presented as figure in Fig.3. Write accordingly
Author response: We have now write the algorithm of geo-fence estimation in the desired format.
Author action: Algorithm 1 is now presented in textual format at page 7.
Reviewer#1, Comment # 5: The authors should include and discuss other related deep learning architectures utilized in other application for a broader vision. Besides, authors should cite more related work some examples are given
Author response: We thank the reviewer for highlighting this point. We have updated our related work as per the suggestion of reviewers. However, unfortunately, we could not include some of the mentioned papers as they are not related to face recognition or tracking of persons.
Author action: The related work is updated (page 3).
Reviewer#1, Comment # 6: Clearly mention the research contributions in introduction section.
Author response: We have highlighted in the revised draft the contribution of our paper in introduction.
Author action: The second paragraph in the introduction section on page number from 1 to 2 clearly state the contribution in this paper by highlighting the gap.
Reviewer#1, Comment # 7: Why face fusion is required as proposed in algorithm 2.
Author response: Face detection using Viola-Jones algorithm is usually used with a single cascade and, to the best of our knowledge, fusion of its three cascades (CART, Haar and LBP) in such particular application is not tried yet and our experimentation has proved significant increase in both face count and accuracy. These results are presented in already submitted manuscript both qualitatively (Figure 4) and quantitatively (Figure 10).
Reviewer#1, Comment # 8: Improve the quality of Fig. 7-10.
Author response: We have now updated the visibility of figure 7, 8,10, 11.
Author action: Updated figures 7,8,10 and 11. (because we have added one more figure which is now Figure 9, therefore figures numbers are now updated after figure 9).
Reviewer#1, Comment # 9: Compare the proposed work with other state-of-the-art works in the same field.
Author response: We are very thankful to the reviewer for in depth analysis of our manuscript. The comparison with state-of-the-art is already presented through table 1 of submitted manuscript. However, we have further added more recent related work in the table 1 as desired by the reviewer (page 3)
Author action: Table 1 is updated with more recent related work (page number 3)
Reviewer 2 Report
The paper is nicely written but I have the following comments:
11. Why the authors called the mosque (Al Nabvi Mosque, Madinah, KSA). It should be called using its native Arabic name Al-Nabawi Mosque.
22. it’s a smart system --> it is (more formal)
33. Assume there are thousands of people visiting the holy mosque during Hajj and a person got lost. How can you extract the face of the lost person from hours of recorded video? In figure 1, it says “missed person’s face image,” how did you get this?
44. How is the proposed system compared to alternative systems such as placing tracking devices on elderly people or children? How is more effective?
55. How can you generalize the proposed model from Almasjid AlNabawi to a general public place?
66. Please add a comparison with previous studies
Author Response
Reviewers Comments, Authors’ Reponses and Actions
Reviewer # 2
Reviewer#2, Comment # 1: Why the authors called the mosque (Al Nabvi Mosque, Madinah, KSA). It should be called using its native Arabic name Al-Nabawi Mosque.
Author response: We have now replaced Al Nabvi mosque with “Al Nabawi Mosque”.
Author Action: We have now replaced all the occurrences of Al Nabvi mosque with “Al Nabawi Mosque” in the revised draft.
Reviewer#2, Comment # 2: it’s a smart system --> it is (more formal)
Author response: We have now address this point of reviewer and removed “smart” as it is an automated system.
Author action: We have now replaced smart system with automated system in the second last paragraph of section 1 Introduction on (page number 2).
Reviewer#2, Comment # 3: Assume there are thousands of people visiting the holy mosque during Hajj and a person got lost. How can you extract the face of the lost person from hours of recorded video? In figure 1, it says “missed person’s face image,” how did you get this?.
Author response: It is illustrated in Figure 1 that when a person is lost the group leader or caretaker will report the missing person online along with the location and time. The system will have profile images of all the pilgrims in the database which will be used to track them as shown in figure 1 and explained in section 3.1.
Reviewer#2, Comment # 4: How is the proposed system compared to alternative systems such as placing tracking devices on elderly people or children? How is more effective?
Author response: There are several other methodologies that are applied for tracking persons in similar applications. However, our work is focused on Al Nabawi mosque and we have to utilize the existing infrastructure of installed cameras there in order to investigate the tracking of missing persons. However, there are several other methodologies and research dimensions like “gait recognition”, “person re-identification” and “Tracking using wearable devices” that need to be explored and are part of our planned future work.
Author action: We have updated our future work in response to the highlighted point of the reviewer on page number 22.
Reviewer#2, Comment # 5: How can you generalize the proposed model from Almasjid AlNabawi to general public place?
Author response: The proposed model consists of two main modules. The first module reduces search space by geofence set estimation based upon missing person’s reported approximate missing location and time lapsed since he missed. Then the second module does automated face recognition of that missed person in videos of installed cameras with in the premises of all geofences that belong to the estimated set of geofences. So, in general, this model is applicable for automated video surveillance of any large crowd gathering public place which is divided into equally square sized geofences and where immediate reporting of missed person is done.
Reviewer#2, Comment # 6: Please add a comparison with previous studies?
Author response: We are very thankful to the reviewer for in depth analysis of our manuscript. The comparison with state-of-the-art is already presented through table 1 of submitted manuscript. However, we have further added more recent related work in the table 1 as desired by the reviewer (page 3).
Author action: Table 1 is updated with more recent related work (page number 3).
Reviewer 3 Report
This paper presents a system to report missing persons and automatically geo-fence the most relevant regions of a mosque yard, and then proceed to automatically detect faces in the streams of cameras located in selected regions, to ultimately identify the missing person.
Geo-fencing and "narrowing down" of the problem is presented as novel, although it has been used extensively in the past.
Face detection is performed via Viola Jones and cascades (HAAR, etc.) which is also not novel. This detection is applied specifically on a single video feed of 2208 frames from what seems to be a single camera.
Finally, face recognition is performed from a total of 11 individuals, that have been previously detected. However, in the video there are many more individuals whose face is not showing frontally, and the algorithm fails to detect these faces. Therefore, the usability/practicality of the proposed system is seriously compromised.
Furthermore, the majority of detected faces are of individuals not wearing kufya or other turbans. These elements add additional dificulty to face detection, and have not been considered in the work. The same applies to women wearing hijab/al-amira/etc. Since most can be similar in colour and textures, this would add additional difficulty for recognition. However, this does not seem to have been studied.
Finally, the conclusions are not supported by the results. That is, the goal of identifying missing people is not achieved successfully, as the experiments have been very limited in their scope (number of people, number of cameras, number of videos, variability of dressing and gender, etc.)
Author Response
Reviewers Comments, Authors’ Reponses and Actions
Reviewer # 3
Reviewer#3, Comment # 1: 3.1 Geo-fencing and "narrowing down" of the problem is presented as novel, although it has been used extensively in the past.
Author response: Dear reviewer, the proposed model consists of two main modules. The first module reduces search space by geofence set estimation based upon missing person’s reported approximate missing location and time lapsed since he missed. Then the second module does automated face recognition of that missed person in videos of installed cameras with in the premises of all geofences that belong to the estimated set of geofences. So, in general, this model is applicable for automated video surveillance of any large crowd gathering public place which is divided into equally square sized geofences and where immediate reporting of missed person is done.
To the best of our knowledge, dividing the surveillance premises into geofences and estimating set of geofences for probable presence of missing person is used for the first time in this work. However, it will be very helpful for us if honorable reviewer provide references of automated video surveillance works that are applied exactly in this way.
Reviewer#3, Comment # 2: Part A: Face detection is performed via Viola Jones and cascades (HAAR, etc.) which is also not novel.
Part B: This detection is applied specifically on a single video feed of 2208 frames from what seems to be a single camera
Author response Part A: Dear reviewer, you are right that Viola-Jones algorithm is the most famous algorithm that is used in such real time applications for face detection. But it is usually used with a single cascade and, to the best of our knowledge, fusion of its three cascades (CART, Haar and LBP) in such particular application is not tried yet and our experimentation has proved significant increase in both face count and accuracy. These results are presented in already submitted manuscript both qualitatively (Figure 4) and quantitatively (Figure 10)
Author response Part B: We think that there is some misunderstanding regarding number of cameras used to collect the dataset of our experimentation. Dear reviewer, we have mentioned in already submitted manuscript that there are 20 cameras that are installed in Al Nabwi mosque and the dataset that we used in our experimentation consists of 2208 images that are collected through installed cameras in the mosque. However, we are very thankful to reviewer for pointing out this aspect and to overcome such confusion for general reader we have included another figure (i.e. Figure 8) in updated manuscript.
Author action Part A: No action required
Author action Part B: Additional figure 9 that clearly describe the data collection from multiple cameras has been included in section 4 on page number 13.
Reviewer# 3, Comment # 3: Finally, face recognition is performed from a total of 11 individuals, that have been previously detected. However, in the video there are many more individuals whose face is not showing frontally, and the algorithm fails to detect these faces. Therefore, the usability/practicality of the proposed system is seriously compromised
Author response: For face detection of a person, it is necessary that a person’s face is visible in front of camera. Furthermore, there are several more factors for proper face detection like image resolution. So in low resolution large crowd gathering images, it is not possible to detect all individual’s faces. Our detection algorithm even increases the accuracy of faces detection in such images as it detects faces by fusion of three cascades of Voila Jones algorithm. The results are presented in already submitted manuscript both qualitatively (Figure 4) and quantitatively (Figure 10)
Reviewer# 3, Comment # 4: Furthermore, the majority of detected faces are of individuals not wearing kufya or other turbans. These elements add additional dificulty to face detection, and have not been considered in the work. The same applies to women wearing hijab/al-amira/etc. Since most can be similar in colour and textures, this would add additional difficulty for recognition. However, this does not seem to have been studied
Author response: Dear reviewer, this research work is focused on tracking missing person through face recognition in videos and offcourse it is not possible when faces are covered. So, in order to cover other dimensions like you mentioned, we have planned to work in future in the research field of “gait recognition”, “person re-identification” and “Tracking using wearable devices”. So these are the part of our future work and this work is limited to face recognition.
Author Action: We have updated the last section “conclusion and future work” in response to the highlighted point of the reviewer in the revised manuscript on page number 22.
Reviewer#3, Comment # 5: Finally, the conclusions are not supported by the results. That is, the goal of identifying missing people is not achieved successfully, as the experiments have been very limited in their scope (number of people, number of cameras, number of videos, variability of dressing and gender, etc.)
Author response: This research work is focused on tracking missing person in Al-Nabwi mosque using existing infrastructure which consists of already installed cameras. So it focused on “face recognition” in those cameras videos. So, in order to cover other dimensions like detecting missing person when his/her face is hidden, more research is required in other research fields like “gait recognition”, “gender classification”, “person re-identification” and “Tracking using wearable devices” that are the part of our planned future work.
Author action: The last section of “conclusion and future work” is updated in order to accommodate this explanation on page number 22.
Round 2
Reviewer 2 Report
Authors have addressed my comments
Author Response
We are thankful to the reviewer for accepting our revision.
Reviewer 3 Report
I stand by my initial decision regarding this work: it provides insufficient experimental results. All results of figures 3--5, and 12 are of the same camera view, regardless that the new Fig.9 includes information about the cameras where different clips come from (I appreciate the authors including this, as it was truly necessary to better understand their work). All face detection results shown are of the same camera view (Figs. 3--5, 12). In my opinion, there should be more views (from different cameras) of the same subjects that have moved among different camera views (tracking, re-detection). One can observe that in all examples shown, the subjects are sitting. These do not seem to be good examples of face re-detection in real-life situations to find missing people, where the person has displaced from the field of view of one camera to another. I think further examples, and further experiments would be necessary in order to be able to conclude that the system works for re-detection. The authors should make an effort to clearly show re-detection among more different camera views. From the current presentation of the paper, this is not clear.
An idea/suggestion: with regard to traditional/religious head coverings, these could be included as part of a "head model" in which both the face and the covering are modelled into a single "subject head model" that includes "front", "sides" and "back" information (in a 3D fashion). This additional information might be helpful to provide re-detection of similar-looking people by integrating more information of their clothing. The "smoothness" assumption might be exploited to add more images to a head model based on the region where the same head was detected in previous frames.
Author Response
Reviewer 3 Comment 1: I stand by my initial decision regarding this work: it provides insufficient experimental results. All results of figures 3--5, and 12 are of the same camera view, regardless that the new Fig.9 includes information about the cameras where different clips come from (I appreciate the authors including this, as it was truly necessary to better understand their work). All face detection results shown are of the same camera view (Figs. 3--5, 12). In my opinion, there should be more views (from different cameras) of the same subjects that have moved among different camera views (tracking, re-detection). One can observe that in all examples shown, the subjects are sitting. These do not seem to be good examples of face re-detection in real-life situations to find missing people, where the person has displaced from the field of view of one camera to another. I think further examples, and further experiments would be necessary in order to be able to conclude that the system works for re-detection. The authors should make an effort to clearly show re-detection among more different camera views. From the current presentation of the paper, this is not clear.
Author Response: We are thankful to reviewer for suggesting tracking of same person in different camera views. We have added Figure 6 in section 3.5 (i.e Missed Person Tracking) that shows the tracking of some subjects in different camera views. We feel that the revised manuscript has improved and we are really thankful for the reviewer for diverting our attention towards this important aspect.
Author Action: We have added Figure 6 in section 3.5 (page no 11 & 12) that shows the tracking of some subjects in different camera views.
Reviewer 3 Comment 2: An idea/suggestion: with regard to traditional/religious head coverings, these could be included as part of a "head model" in which both the face and the covering are modelled into a single "subject head model" that includes "front", "sides" and "back" information (in a 3D fashion). This additional information might be helpful to provide re-detection of similar-looking people by integrating more information of their clothing. The "smoothness" assumption might be exploited to add more images to a head model based on the region where the same head was detected in previous frames.
Author Response: Dear reviewer this work is focused on tracking through face recognition in low resolution images only. Your suggestion of implementing 3D head modelling is fantastic, however it can be implemented for high resolution images only and it’s a good suggestion for the future work.
